# CRISPR analysis suggests that small circular single-stranded DNA smacoviruses infect *Archaea* instead of humans

César Díez-Villaseñor [1] & Francisco Rodriguez-Valera [1]

*Smacoviridae* is a family of small (~2.5 Kb) CRESS-DNA (Circular Rep Encoding Single-Stranded (ss) DNA) viruses. These viruses have been found in faeces, were thought to infect eukaryotes and are suspected to cause gastrointestinal disease in humans. CRISPR-Cas systems are adaptive immune systems in prokaryotes, wherein snippets of genomes from invaders are stored as spacers that are interspersed between a repeated CRISPR sequence. Here we report several spacer sequences in the faecal archaeon *Candidatus* Methanomassiliicoccus intestinalis matching smacoviruses, implicating the archaeon as a firm candidate for a host. This finding may be relevant to understanding the potential origin of smacovirus-associated human diseases. Our results support that CRESS-DNA viruses can infect non-eukaryotes, which would mean that smacoviruses are the viruses with the smallest genomes to infect prokaryotes known to date. A probable target strand bias suggests that, in addition to double-stranded DNA, the CRISPR-Cas system can target ssDNA.

[1] Evolutionary Genomics Group, Departamento de Producción Vegetal y Microbiología, Universidad Miguel Hernández, San Juan de Alicante 03550, Spain. Correspondence and requests for materials should be addressed to C.D-V. (email: cesardiezvillasenor@outlook.com)

Viruses with a single-stranded (ss) genome comprise Group II of the Baltimore classification. Together with RNA viruses, ssDNA viruses include the smallest known viruses and are prone to mutations and recombination[1,2]. Both viral groups comprise huge diversity and numbers previously underestimated, which are currently being uncovered[3]. According to the classification system specified by the International Committee on Taxonomy of Viruses (ICTVs) (https://talk.ictvonline.org/taxonomy/), there are 13 accepted families of ssDNA viruses, which infect hosts in all three domains of life. Circular Rep Encoding Single-Stranded DNA (CRESS-DNA) viruses are thought to infect eukaryotes. Of the six official families of CRESS-DNA viruses (more have been proposed)[2], there are five with isolates (cultivated) known to infect plants (*Geminiviridae*[4], *Nanoviridae*[5]), vertebrates (*Circoviridae*[6]), diatoms (*Bacilladnaviridae*[7]) or fungi and vector insects[8] (*Genomoviridae*[9]). The genomes of most CRESS-DNA families are monopartite, 2–3 kb in length, and encode for Rep (a protein involved in rolling circle replication, which shares a common origin among CRESS-DNA viruses) and a capsid protein (the capsid is icosahedral). Replication begins at a DNA stem-loop structure after the genome is converted to double-stranded DNA (dsDNA) by host factors[10]. Most CRESS-DNA viruses remain uncultured, as their hosts are unknown[1,2]. Numerous metagenomic studies have demonstrated an impressive diversity of Rep proteins encoded by CRESS-DNA viruses, which appear to be widespread in numerous habitats[1–3]. One example, which is found only in faecal metagenomes (from insects and vertebrates), are smacoviruses (small, circular genome viruses)[1], which constitute the CRESS-DNA family *Smacoviridae*[11]. The *Smacoviridae* have genomes of approximately 2.5 kb, which encode only Rep and a capsid protein (Cap). Although these viruses are thought to infect eukaryotes, their actual host has not yet been confirmed.

In prokaryotes, one method for assigning a virus to a host takes advantage of CRISPR (Clustered Interspaced Short Palindromic Repeats)[12,13]. CRISPR-Cas systems[14,15] consist of arrays of CRISPR and its associated Cas (CRISPR-associated) proteins[16]. Interspersed among the repeats are distinct (un-repeated) spacer sequences. Spacers derive from sequences (protospacers) in mobile genetic elements, such as viruses and plasmids that use the CRISPR-containing prokaryote as a host[17–20], and spacers are primarily involved in adaptive immunity against the invading sequences by facilitating the recognition of such sequences[17–19,21]. Processed CRISPR RNA (crRNA) binds Cas proteins to form an interference complex that recognizes target sequences through base paring.

The origins of CRISPR spacers make it possible to assign virus–host pairs in prokaryotes in a highly reliable way when there is an identical match between a spacer and a protospacer[22]. The observation that prompted this work was the serendipitous discovery of matches (one perfect match) between spacers of the CRISPR subtype I-B in the methanogenic archaeon *Candidatus Methanomassiliicoccus intestinalis* Issoire-Mx1, accession number NC_02135326, and a smacovirus, accession number NC_0262523. The virus was described as having *Homo sapiens* as a host and was found, like other smacoviruses, in faecal samples from humans with unexplained gastroenteritis. However, whether smacoviruses were the cause of the disease remained unclear, as did the identity of the viral host.

The predicted host, *Ca*. M. intestinalis, belongs to the recently described order Methanomassiliicoccales. The metabolic hallmark of this order is the dependence on an $H_2$ source to reduce methyl compounds to methane[23–25]. Methanomassiliicoccales are present in gastrointestinal tracts (GITs) and environmental samples[26] (primarily in wetlands). Their wide distribution and abundance suggest that the order plays an important role in

methane emissions and global warming[26]. *Candidatus* M. intestinalis has been obtained in a highly enriched culture from human faeces[27]. Our current knowledge of the species derives from the analysis of its genome and its growth in an enrichment culture; its prevalence is unknown.

We report also additional similarities between smacoviruses and the spacers of *Ca*. M. intestinalis, which are analysed here. Our results indicate that CRESS-DNA viruses could also infect prokaryotes. Although infection and viral autonomy should be empirically confirmed, this finding could mean the identification of the smallest prokaryotic viruses known to date.

## Results

**CRISPR of *Ca*. M. intestinalis targets *Smacoviridae*.** Prior to this study, we performed a search of viral genomes for sequences similar to CRISPR spacers (see 'Previous BLASTn search' in the Methods section), which should relate viruses and their hosts. For RefSeq viruses their hosts were already described, but as previously noticed[22] there were a few discrepancies at the domain level. A closer look at the CRISPR-predicted hosts with described hosts from other domains uncovered one unambiguous case that stood out: several spacers from the archaeon *Ca*. M. intestinalis (NC_021353) matched protospacers in a human smacovirus (NC_026252, KP264966). This case was exceptional because there were more than one matching spacer highly similar to their putative protospacers. Remarkably, the described viral host of the smacovirus was *H. sapiens*, and all known hosts of CRESS-DNA viruses are eukaryotes.

*Candidatus* M. intestinalis Issoire-Mx1 has a previously described[24] complete subtype I-B CRISPR-Cas system (Fig. 1, top) with 112 repeats (array B1, of which the repeated sequence is 5′-gttagaaatccatctaaactagaatgtaaat-3′, palindromic sequences are underlined). All Cas proteins are of the subtype I-B except for Cas8, which belongs to the subtype I-A (Cas8a1)[24]. We also noticed that a previously described small cluster of three repeats (array B2), approximately 3 kb from the array B1, shares the same repeated sequence in the complementary strand. For both arrays, a so-called leader sequence[16,19] can be identified on the basis that the most distant repeat has a poorly conserved sequence[16]. As CRISPR promoters are in the leader sequence[28], transcription would be the same as predicted previously[24]. As expected, both leader sequences are AT rich[13] but have no significant identity. Interestingly, the only two genes predicted to have similarities with viruses in the genome[24] are located between arrays B1 and B2.

A specific search against a previously published[11] selection of 83 smacovirus sequences revealed similarities to 23 spacers of *Ca*. M. intestinalis (Fig. 1; Table 1). Such a high number of spacers may be needed to stop viral infection due to high sequence diversity or because spacers farther from the leader end are poorly expressed. Multiple targets may also be necessary to abort an infection if only double-stranded (ds) DNA is cleaved. For this search, local regions of the subject sequences were aligned against the entire length of spacers (we call this algorithm local–global alignment, see Methods for details). Matches for four additional spacers were found by local alignments, but were not considered in this work unless otherwise stated. Representatives of best alignment to each spacer are shown in Supplementary Table 1. All spacers with targets belong to the B1 array and are identified by their distance from the leader preceded by sp (spacer) or psp (protospacer) for targets (all targets are putative).

**Validity of the virus–host prediction.** Although the cut-off *e*-value used was 0.01, only one best match to a spacer had an *e* value under $3 \times 10^{-4}$, and 18 of them were under $10^{-4}$, which is

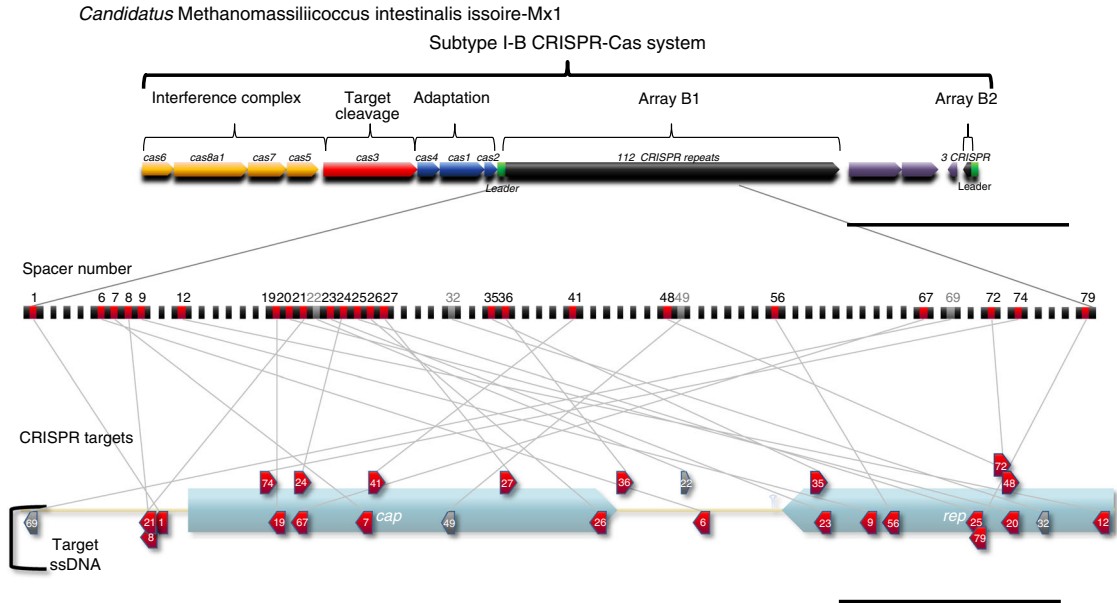

**Fig. 1** Clustered Interspaced Short Palindromic Repeats (CRISPR)-Cas Subtype I-B in *Candidatus* M. intestinalis and its targets in smacoviruses. The CRISPR-Cas subtype I-B in *Ca*. M. intestinalis is represented. Related *cas* genes are grouped and coloured according to functional categories: adaptation (*cas1–2* and *cas4*), target cleavage (*cas3*) and formation of interference complex (*cas6, cas8a1, cas7* and *cas5*). Cas8a1 would belong to subtype I-A. Two repetitive arrays are found in opposite orientations, here named B1 (112 repeats) and B2 (3 repeats), with putative leader sequences, from where transcription would start. Array B1 is amplified and similarities of spacers to sequences in smacoviruses indicated. A smacovirus 'archetype' is depicted where targets are represented with their relative positions to genes (see Methods). Similar sequences that are in the same orientation of spacers point right and vice versa. Smacoviruses are circular single-stranded DNA (ssDNA) virus with just two genes (*rep* and *cap*). Targets in the ssDNA are represented below the viral genome, and targets in the complementary strand above. Targets in red were found with local–global alignments, while targets in grey only with local alignments. All arrows point in the sense of predicted transcription (for targets transcription of CRISPR RNA (crRNA)). Scale bars are 5 kb (above) and 0.5 kb (bottom). Non-CRISPR genes are coloured violet

---

### Table 1 Targets of *Ca*. M. intestinalis CRISPR in smacoviral genomes

| Smacoviruses[a] Accession | Species[11] | Country | sp1 | sp6 | sp7 | sp8 | sp9 | sp12 | sp19 | sp20 | sp21 | sp23 | sp24 | sp25 | sp26 | sp27 | sp35 | sp36 | sp41 | sp48 | sp56 | sp67 | sp72 | sp74 | sp79 | Target Group[d] |
|---|---|---|---|---|---|---|---|---|---|---|---|---|---|---|---|---|---|---|---|---|---|---|---|---|---|---|
| **Max identity[c] (%)** (Individual identities[c] below) | | | 89 | 100 | 92 | 87 | 100 | 89 | 86 | 100 | 94 | 100 | 100 | 100 | 78 | 89 | 81 | 76 | 89 | 89 | 83 | 84 | 81 | 78 | 75 | |
| KJ577817[61] | *Porcine associated porprismacovirus 8* | USA | | | | | | | | | | | | | | | | | | | | 78 | | | | Porcine |
| KC545226[61] | *Porcine associated porprismacovirus 2* | USA | | | | | | | | | | | 73 | | | | | | | | | | | | | Porcine |
| KJ577818[61] | *Porcine associated porprismacovirus 2* | USA | | | | | | | | | | | | | | | | | | | 84 | 81 | 78 | | | Porcine |
| KC545230[61] | *Porcine associated porprismacovirus 3* | USA | | | | | | | | | 77 | | | | | | | | | | 76 | | 75 | | | Porcine |
| KC545229[61] | *Porcine associated porprismacovirus 3* | USA | | | | | | | | | 77 | | | | | | | | | | 76 | | 75 | | | Porcine |
| KC545227[61] | *Porcine associated porprismacovirus 3* | USA | | | | | | | | | 77 | | | | | | | | | | 76 | | 75 | | | Porcine |
| KC545228[61] | *Porcine associated porprismacovirus 3* | USA | | | | | | | | | 77 | | | | | | | | | | 73 | | 75 | | | Porcine |
| KM573772[62] | *Camel associated porprismacovirus 1* | United Arab Emirates | | | | | | | | | | | | | | | | | | | | | 75 | | | Mamal |
| KT862225[63] | *Porcine associated porprismacovirus 10* | New Zealand | | | | | | | | | | | | | | | | | | | | | 74 | | | Mamal |
| KP233191[64] | *Gorilla associated porprismacovirus 1* | USA | | | | | | | | | | | 76 | | | | | | | | | | | | | Mamal |
| KP233192[64] | *Gorilla associated porprismacovirus 1* | USA | | | | | | | 74 | | | | 76 | | | | | | | | | | | | | Mamal |
| KM573770[62] | *Camel associated porprismacovirus 2* | United Arab Emirates | | | | | | | | | | | 76 | | | | | | | | 73 | | 78 | | | Mamal |
| KM573775[62] | *Camel associated porprismacovirus 4* | United Arab Emirates | | | | | | | | | | | | | | | | | | | | | | 75 | | Mamal |
| KT862223[63] | *Bovine associated huchismacovirus 1* | New Zealand | | | | | | 81 | | | | | | | | | | | | | | | | | | Mamal |
| KY086298[64] | *Chicken associated porprismacovirus 1* | Brazil | | | | | | | | | | | 76 | | 81 | | | 89 | 83 | | 81 | | | | | Chicken0 |
| KP233178[64] | *Human associated huchismacovirus 3* | France | | 100 | 92 | 84 | 100 | 89 | 80 | 100 | 94 | 73 | | | 78 | 87 | | 89 | | | 76 | | | | | Human A |
| KP233179[1] | *Human associated huchismacovirus 3* | France | | 97 | 92 | 84 | | 89 | 80 | 97 | 94 | 76 | | | 78 | 89 | | 76 | | | | | | | | Human A |
| KP264968[1] | *Human associated huchismacovirus 3* | France | 89 | 97 | 81 | 87 | 100 | 77 | 74 | 84 | 75 | 76 | | | 78 | 84 | 76 | 89 | | | | | | | | Human A |
| KP233177[1] | *Human associated huchismacovirus 2* | France | | 100 | | | 100 | | | 84 | | 73 | | | | | | | | | | | | | | Human B |
| KP233176[1] | *Human associated huchismacovirus 2* | France | | 100 | | | 100 | | | 84 | | 73 | 73 | | | | | | | | | | | | | Human B |
| KP233174[1] | *Human associated huchismacovirus 2* | France | | 97 | | | 100 | | | 87 | | 76 | 73 | | | | | | | | | | | | | Human B |
| KP233175[1] | *Human associated huchismacovirus 2* | France | | 100 | | | 100 | 73 | | 87 | | 76 | | | | | | | | | | | | | | Human B |
| KP264965[1] | *Human associated huchismacovirus 2* | France | | 97 | | | 100 | | | 87 | | | | | | | | | | | | | | | | Human B |
| KP233187[1] | *Human associated huchismacovirus 2* | USA | | 97 | | | | | | 87 | | | 100 | | 100 | | | | | | | | | | | Human B |
| KP233184[1] | *Human associated huchismacovirus 2* | USA | | 100 | | | | | | 84 | | | 100 | | 92 | | | | | | | | | | | Human B |
| KP264967[1] | *Human associated huchismacovirus 2* | France | | 100 | | | | | | 84 | | | 100 | | 92 | | | | | | | | | | | Human B |
| KY086301[64] | *Chicken associated huchismacovirus 1* | Brazil | 73 | | | | 82 | | | 73 | | 81 | 81 | | | | | | | | | | | | | Huchicken |
| KY086300[64] | *Chicken associated huchismacovirus 2* | Brazil | 82 | | | | | | | 81 | | 81 | 76 | | | | | | | | 76 | | | | | Huchicken |
| KY086299[64] | *Human associated huchismacovirus 1* | Brazil | | | | | | | | 76 | 83 | 87 | 100 | 76 | 81 | | | | | | | | | | | Human C |
| KP233188[1] | *Human associated huchismacovirus 1* | USA | | | | | | | | 76 | 77 | 87 | 100 | 95 | 97 | | | | | | | | | | | Human C |
| KP264964[1] | *Human associated huchismacovirus 1* | France | | | | | | | | 76 | 77 | 84 | 97 | 97 | 100 | | | | | | | | | | | Human C |
| KP233186[1] | *Human associated huchismacovirus 1* | USA | | | | | | | | 76 | | 87 | 92 | 100 | 97 | | | | 75 | | | | | | | Human C |
| KP233185[1] | *Human associated huchismacovirus 1* | USA | | | | | | | | 76 | | 87 | 97 | 97 | 100 | | | | | | | | | | | Human C |
| KP264969[1] | *Human associated huchismacovirus 1* | France | | | | | | | | 76 | 80 | 87 | 100 | 95 | 100 | | | | | | | | | | | Human C |
| KP233181[1] | *Human associated huchismacovirus 1* | France | | | | | | | | 76 | 80 | 87 | 97 | 92 | 100 | | | | | | | | | | | Human C |
| KP264966[1] | *Human associated huchismacovirus 1* | France | | | | | | | | 76 | 86 | 84 | 95 | 92 | 100 | | | | | | | | | | | Human C |
| KP233180[1] | *Human associated huchismacovirus 1* | USA | | | | | | | | 74 | | 87 | 95 | 92 | 94 | | | | | | | | | | | Human C |
| KP233182[1] | *Human associated huchismacovirus 1* | USA | | | | | | | | 74 | | 87 | 95 | 92 | 94 | | | | | | | | | | | Human C |
| KP233183[1] | *Human associated huchismacovirus 1* | USA | | | | | | | | 76 | | 87 | 97 | 95 | 100 | | | | | 75 | | | 76 | | | Human C |
| KP233193[1] | *Human associated huchismacovirus 1* | USA | | | | | | | | 76 | | 87 | 97 | 95 | 100 | | | | | | 75 | | 76 | | | Human C |

[a]Genomes of smacovirus with targets, in rows, are sorted according to their position in a published[11] tree based in complete genomic sequences.
[b]Spacers with relevant matches. Vertical lines divides them in tree groups depending on their distance from the leader sequence (see section 'Groups of putatively targeted smacoviruses' in Results for details).
[c]Identity represents alignments of the whole spacer sequence to targets. Identities are highlighted in a greyscale according to their numbers, representing darker tones a higher value.
[d]We defined groups of genomes (target groups) according to their closeness and target profile. Horizontal dashed lines highlight the separation of the groups

an indicator of the correspondence of matches to sequences in *Smacoviridae*. Notably, the redundancy of subject sequences makes these *e* values an overestimation. As previously mentioned, six spacers resulted in 100% identity matches to genomes of smacoviruses. Gaps in the alignments of protein coding regions (Supplementary Table 1) are either compensated between the spacer and target (the same number in each) or, alternatively, the difference of gaps between the query and subject is three (exceptions are sp79 and sp35). This finding implies that indel mutations would not have altered the frame of the coding region, which supports the legitimacy of the alignments. An additional validation of thresholds is that, for spacers from other Methanomassiliicoccales (sequences from Methanomassiliicoccales analysed are listed in Supplementary Table 2, and CRISPR arrays in Supplementary Table 3), we obtained no significant matches to either a set of smacoviral genomes[11] or a set of CRESS-DNA sequences[2]. The only significant hit in these searches was similar to one previously described:[24] a spacer in *Candidatus* Methanomethylophilus alvus targeting a sequence (93% identity) from a ssDNA virus (JX305998). We found, for the same spacer, a similar match (also 93%) to sequence KU203352, which is also classified as a smacovirus[2].

The following trait of protospacers can be used to confirm matches: in Type I CRISPR-Cas systems, one requirement for interference and acquisition is a short motif (PAM, Protospacer Adjacent Motif) next to the protospacer[29], located in the protospacer region corresponding to the upstream zone of spacer transcription (towards the leader). Once protospacers were detected, the best matches to each spacer (Supplementary Table 1) were used to investigate the presence of a PAM. A sequence logo from all six protospacers with 100% identity revealed the conservation of a thymine (Fig. 2b) in the sixth base downstream (downstream or upstream in the protospacers make reference to the spacer transcription); this thymine was not conserved in other protospacers (Fig. 2d). A sequence logo revealed the PAM 'CCN' (Fig. 2a–c) located upstream (bases −3, −2 and −1). This motif was equally conserved when sequences not identical to spacers were accounted for. Most upstream sequences (18 of 23: 78%) present a complete motif, and 100% of the upstream sequences have at least one of the nucleotides; this is a high level of conservation for Type I systems, similar to acquisition studies[30,31], and there is a flexibility in the recognition for interference[32,33]. The presence of PAM sequences in smacoviruses confirms the origin of the spacers and the predicted transcription sense, given that in Type I systems the motifs are located upstream.

There also exists a biological property that relates *Smacoviridae* with their predicted hosts. Methanomassiliicoccales such as *Ca*.

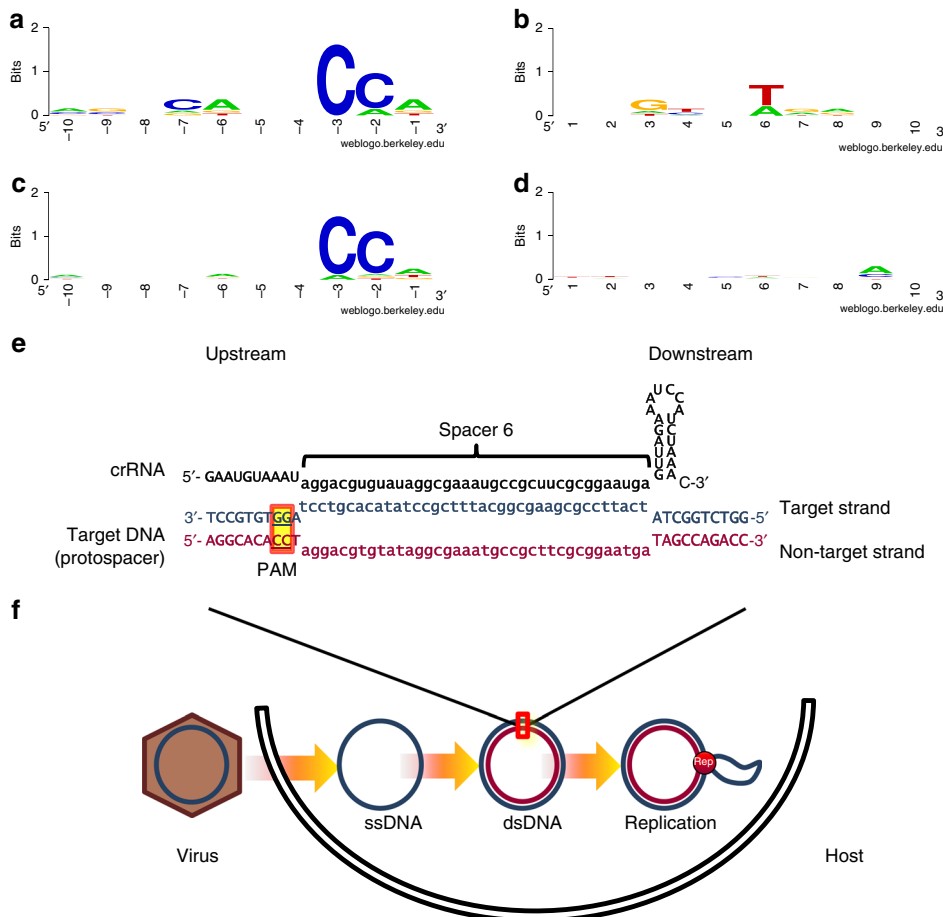

**Fig. 2** Sequence conservation in flanking regions of smacoviral *Candidatus* M. intestinalis Clustered Interspaced Short Palindromic Repeats (CRISPR) targets. Sequence logos represent conservation of 10 bases, in non-target strand, upstream (5' end of corresponding spacer in CRISPR RNA (crRNA)) (**a**, **c**) and downstream (**b**, **d**) from sequences targeted by the spacer in *Ca*. M. intestinalis. Up (**a**, **b**), conservation around the six matches identical to spacers. Down (**c**, **d**), conservation in best matches to each 23 spacers. **e** Representation of an R-loop, where crRNA from spacer 6 hybridizes to target double-stranded DNA (dsDNA) protospacer in smacovirus. **f** Part of life cycle of a Circular Rep Encoding Single-Stranded DNA (CRESS-DNA) virus showcasing presence of dsDNA, targeted presumably by the Type I CRISPR-Cas system, after synthesis of complementary strand, and in replication. In **e**, **f** the viral strand is represented in blue, complimentary in garnet and RNA in black

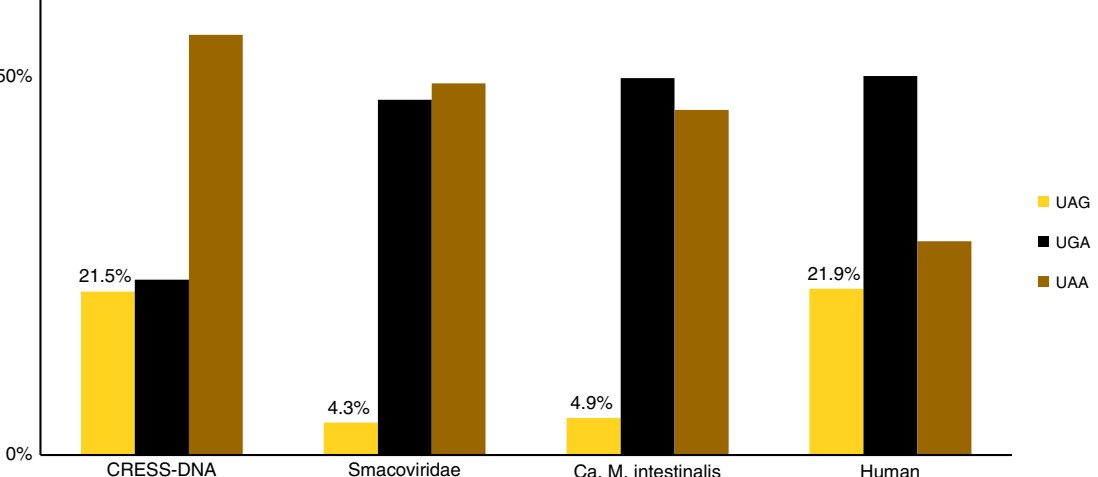

**Fig. 3** Stop codons usage of *Smacoviridae* compared to Circular Rep Encoding Single-Stranded DNA (CRESS-DNA) viruses and possible hosts. In organisms, such as *Candidatus* M. intestinalis, that use *amber* (UAG) codon to codify pyrrolysine, its use for translation termination is reduced. Bars represent the percentage of coding sequences ending in one of the standard termination codons that end with *amber*, *opal* (UGA) or *ochre* (UAA) codons for a set of CRESS-DNA virus, a set of smacoviruses, *Ca.* M. intestinalis genome and human genome (see details in Methods section). Only percentages of amber codon are shown

M. intestinalis adopt the *amber* stop codon ('UAG') to encode the amino acid pyrrolysine; as a result, the coding sequences of these archaea are biased against the use of the *amber* stop codon for translation termination[24]. The percentage of each stop codon in the predicted proteins (188) of the set of 83 smacovirus genomes was analysed, and the *amber* codon was found to be utilized in only 4.25% of coding sequences. Notably, the use of termination codons in smacoviruses is similar to that of *Ca.* M. intestinalis[24] but different from that of human or CRESS-DNA coding sequences (Fig. 3). None of the viruses with *amber* stop codons (GQ351273, GQ351276, GQ351277, KP233190, KT862219, KT862222, KX838318) include any detected *Ca.* M. intestinalis CRISPR targets.

**Distribution of putative targets and matching spacers**. To visualize the positions of the targets, an archetype smacovirus depicting all protospacers was drawn (Fig. 1, bottom). The targets were distributed along the entire archetype, with no region excluded. There was a slightly higher density of targets in the *rep* gene than in *cap* (10 and 9, respectively, with *cap* being 30% larger), possibly caused by a higher conservation of the *rep* gene. There was also a slight bias towards targeting the template strand (7 coding vs. 12 template).

Even in a ssDNA virus, targets can be located in either strand. This is not surprising given that, for replication, smacoviruses must form dsDNA, and new spacers are incorporated from dsDNA[34]. CRESS-DNA viruses enter the cell as ssDNA (this strand will be referred to as the 'viral strand'), and then host factors synthesize the complementary strand, which is used as a template for rolling circle replication, yielding multiple circular single-stranded copies of the viral genome. Although both strands are targeted, we detected a bias favouring the viral ssDNA; only 8 of the 23 spacers were found to target the complementary strand (35%). According to the binomial distribution, the probability of obtaining this bias or higher towards targeting the viral strand by chance is 0.105 (if matches found only in local alignments are included, the probability is reduced to 0.061). This bias suggests that the present CRISPR-Cas system affects target sequences in ssDNA in addition to targets in dsDNA. This finding would be surprising, as tested Type I systems are known to target dsDNA only[32].

The presence of the targets in each smacovirus and their identity with each spacer are depicted in Table 1. Matches to spacers were detected in multiple smacoviruses, primarily in a subset of related smacoviruses associated with humans (genus *Huchismacovirus*). KP264968 (human-associated) is the viral sequence with the most protospacers detected (14 out of 23 detected). No single virus contained all protospacers (Table 1), which indicates that the protospacers derive from diverse viral genomes. The targets were present in many genomes, or their identity to the spacer was under 90%, and thus the donor of the original protospacer could not be ascertained. There were six spacers (sp1, sp35, sp36, sp48, sp56 and sp79) with targets present in only one of the viral genomes; three of the spacers (sp35, sp48 and sp56) had targets in KY086298 (chicken-associated smacovirus), two of the spacers (sp1 and sp36) had targets in KP264968 (human-associated) and one spacer (sp79) had a target in KM573770 (camel-associated). The genome with the most targets (14) was KP264968. Spacers sp20 and sp23 had targets in the highest number of genomes (26 and 25, respectively), located in the *rep* gene. In *Ca.* M. intestinalis, there were no self-targeting spacers. The only similarities to the archaeon genome were between sp59 and sp63, which are almost identical (one mismatch and sp59 has an extra nucleotide), and between sp34 and sp49, which overlap 19 nucleotides (out of 36 each) in opposite orientations.

Given that spacers are usually incorporated at the leader end of the array, spacers closer to this end are more recently acquired in the genome. As expected[35,36], the putatively targeting spacers were predominantly among the most recent; no similarities were found for the last 41 spacers (out of 111), and most matching spacers were near the leader (14 of the 27 first spacers in B1 array were targeting smacoviruses). In addition, the 25 first spacers were found to include all six that were identical to one target. For the first 25 spacers the lowest identity value was 85.7% (average 94,7%), while the rest of targeting spacers had identity values of 77.8 to 88.9% (average 82.3%). This finding can be explained by a loss of sequence identity over time (more ancient spacers have lower identity). There were also consecutive runs of spacers with targets in smacoviruses (sp6–9, sp19–21, sp23–27 and sp35–36) and blocks with an absence of targets, which suggests periods of fluctuation in exposure to different viruses.

**Groups of putatively targeted smacoviruses**. We defined the following seven groups of targeted smacoviruses according to the relationships among the viruses and the profile of their putative targets: Porcine, Mammal, Chicken0, Human A, Human B, Huchicken and Human C (see Table 1). These names describe the vertebrates from which the viruses were sequenced. The correspondence of target groups to taxonomy[11] is also described in Table 1. It is worth noting that the genus *Huchismacovirus* comprises the groups with the most similar matches to spacers.

Although the timescale is unknown, it is possible to evaluate the history of infections to *Ca*. M. intestinalis Issoire-Mx1 by these different groups of viral genomes, based on the age of the spacers that target them[37]. The following three ages were selected (see vertical lines in Table 1): modern age (sp1–9), middle age (sp12–27) and ancient age (sp35–72). In the modern age, Human A appears to be the principal smacoviruses putatively infecting this archaeon, whereas for the middle age the principal groups are Human A and C. For the ancient age, the spacers are from Human A, a more ancient virus with a *rep* gene similar to that of Chicken0, and from the Porcine group. This finding could either reflect a progressive specialization to the human gut or a degeneration of targeted protospacers to evade immunity. Spacer matches in the groups Huchicken and Mammal appear to be a consequence of homology. The only exception may be psp79, which is found only in the Mammal group.

Protospacers in the more divergent intergenic regions (psp21, psp8, psp1 and psp36, psp6) are found almost exclusively in the Human A group (except for psp6, also found in Human B), which indicates that Human A group are the principal putatively infecting smacoviruses. All smacoviruses from the groups Human A, B and C, which appear to contain the original protospacers (protospacer with maximum identity to spacer being at least 90%), were sequenced from human samples. Therefore, there seems to be a strict association between the archaeal strain, its host, and the virus. Identification and sequencing of additional *Ca*. M. intestinalis strains from other hosts would be needed to confirm such specificity.

## Discussion

Smacoviruses are present in the GIT of humans and healthy animals. Although smacoviruses are suspected to infect eukaryotes, this hypothesis has not been confirmed. The prediction of a eukaryotic host is based on the relationship of smacoviruses with other CRESS-DNA viruses and the presence of an intron in the *rep* gene of a howler monkey-associated smacovirus[1]. A eukaryotic host could be either a component of food or the enteric cells themselves. Two studies argue against a dietary origin: food for cows bearing smacovirus in their stools was shown to be free of viruses[38], and only a fraction of pigs fed with exactly the same diet had smacoviruses[39]. Thus, the host cells of smacoviruses could be the GIT cells, but inoculated immuno-deficient mice failed to replicate the virus[1], which argues against this hypothesis. However, the latter observation in mice could also be explained due to host specificity, as the viruses were isolated from humans. Nonetheless, a prokaryotic host living in the GIT is compatible with these findings, as the prokaryotic host is not a primary component of food, and its presence would be necessary for viral replication. The prediction of a eukaryotic host based on the presence of an intron is also inconclusive due to the existence of archaeal introns[40]. The following factors strongly indicate that smacoviruses infect the archaeon: the finding of several spacers from *Ca*. M. intestinalis Issoire-Mx1 that match smacoviruses, essentially from the same genus (*Huchismacovirus*); both the virus and the predicted host are present in human GITs; the confirmation of a PAM sequence; and the usage of the *amber* stop

codon. Future experiments could confirm whether smacoviruses can replicate in *Ca*. M. intestinalis.

The spacers of *Ca*. M. intestinalis had no similarities to any other CRESS-DNA sequences. Other tested spacers from Methanomassiliicoccales had no significant matches against any of those groups, with the already described exception of a spacer in *Ca*. M. alvus, which matched to a genome that can also be classified as a smacovirus. This finding suggests that smacoviruses may infect other gastrointestinal Methanomassiliicoccales. Whether these virus–host pairs are also found in the environment (not in GITs) remains unknown.

CRESS-DNA-like viral particles with no DNA injection mechanisms may be unable to access the cytoplasm of most prokaryotes. This problem would be easier to overcome in organisms without a cell wall. The cell structure of *M. luminyensis* consists of a double membrane but no cell wall or S layer; these cells are so weak that they lyse in hypotonic distilled water[41]. Some authors generalize this structure to the entire order[26]. In any case, the content of genes that could potentially contribute to a cell wall is equivalent in *Ca*. M. intestinalis and *M. luminyensis*[25].

A pathogenic role of smacoviruses was proposed, as they were discovered in metagenomic analyses of faecal samples from gastroenteritis outbreaks that tested negative for other known viral pathogens but had also been found in healthy animals[1]. Additionally, smacoviruses are distantly related to other CRESS-DNA viruses that are pathogens that infect plants and animals. However, in the present case, Methanomassiliicoccales could be the organisms implicated in pathogenesis. As happens with other methanogens, their implication in disease could be indirectly favouring other microbial pathogens[42,43]. In addition, a therapeutic use of Methanomassiliicoccales has been proposed for the elimination of trimethylamine, which causes thrimethylaminuria[44] and atherosclerosis[43], for which viral infection should be taken into account.

The interference complexes of Type I systems bind targets in both ssDNA[45–47] and ssRNA[48,49] without triggering nuclease activity. The binding of the interference complex to mRNA may neutralize its immune effectiveness, and it has been observed that targets in the coding strand are less effective[50]. Therefore, the interference effectiveness could explain the bias in favour of targeting the template strand. A larger bias was observed towards targeting the viral strand (the actual ssDNA). Type I CRISPR-Cas systems are known to detect targets only in dsDNA. Consequently, for conjugative plasmids, no benefit has been observed from targeting the ssDNA penetrating the recipient to prevent transference[51], which is in contrast with the bias found here. When bound to ssDNA, the interference complex could block the viral cycle with no further action; on the other hand, this binding would render the interference complex ineffective, which could be detrimental to the host. A peculiarity of the CRISPR-Cas system of *Ca*. M. intestinalis is the change of Cas8b to Cas8a1[24], which is the protein that recognizes dsDNA-PAM and subsequently recruits and activates the Cas3 nuclease function[32] to degrade targets. This peculiar arrangement might have created a combination with the capability of recognizing PAMs also in ssDNA. If that were the case, this would be the first known example of a Type I CRISPR-Cas system triggered after the recognition of dsDNA and ssDNA targets, which has only recently been observed for CRISPR-Cas14 ribonucleoproteins[52].

One of the lingering questions about CRISPR spacers is why the targets to most of them remain undetected[18,20]. A recent study revealed that spacers with unknown origins have the same sequence properties as spacers with known origins, which strongly suggests that all spacers originate from species-specific mobile genetic elements[20]. Non-mutually exclusive options

explaining the low frequency of spacers with known protospacers include sequence modifications and our incomplete knowledge about prokaryotic-related elements. The sequence modification of protospacers may be the result of a survivor bias, which is in agreement with a previously observed degeneration of CRISPR viral targets[37,53]. The profile of different identities to the same protospacers observed in this work demonstrates how the variability in target sequences can make them undetectable. It is obvious (see Table 1) that the sensitivity to detect protospacers does not allow for the detection of the same match in the same gene in a different genome. This is because spacers are short sequences and, compared to genes, at the same level of nucleotide identity there is a higher probability of obtaining the same matches by chance. Indeed, the search performed here in a smaller dataset revealed otherwise overlooked protospacers confirmed by PAM sequences. Hence, it is likely that sequences related to many original spacer precursors are known, but it is not possible to trace them. This would be especially noteworthy for viruses with high mutation rates, such as ssDNA viruses. For instance, in *Ca.* M. intestinalis, it is likely that the proportion of older spacers originating from smacovirus has been underestimated. However, none of the matches presented in this paper had been reported in a previous study[24], which indicates that the increase of sequences in the databases is still providing novel information. Data from seminal studies[17–19] to more recent studies[20] continue to confirm our blind spots regarding microbial life. Therefore, this work highlights the importance of re-evaluating our current knowledge as new biodiversity is discovered. A two-step approach to finding statistically significant protospacers, first in a large database and second in sequences related to identified matches, has been proven effective in this case.

The smallest viral genome thought to infect prokaryotes is the ssRNA plasmid-dependent enterobacterial phage M[54] (3405 nucleotides). Accordingly, smacoviruses could be the smallest viral genomes to infect prokaryotes discovered to date. The infection of archaea by CRESS-DNA viruses, which normally infect eukaryotes, would also represent another connection between these two cellular domains.

## Methods

**CRISPR detection**. An evolution of a programme to detect CRISPR, employed in other publications[12,18,29], was used to detect CRISPR arrays. Spacers of arrays with at least four repeats were selected. In certain cases, smaller arrays close to previously selected ones with the same CRISPR sequence were also chosen.

**Previous BLASTn search**. Initially, a standalone BLASTn[55] search for spacers was performed with the following command line modified from other work:[22] "*blastn -task blastn-short -db subjectfilename -query queryfilename -out blastn1out2 -outfmt '6 std qcovs' -evalue 0.003 -word_size 7 -gapopen 10 -gapextend 2 -penalty −1 -max_target_seqs 1000.*" The size of the database is relevant to assess the quality of findings. Queries where non-redundant spacers (390,291 sequences, $1.32502 \times 10^7$ nucleotides) from different prokaryotes, subjects were 195,696 viral genomes ($3.27665 \times 10^9$ nucleotides). Four spacers from *Ca.* M. intestinalis had the following matches with a smacovirus (*e* values within parentheses): sp20 (0.002), sp23 ($7.48 \times 10^{-7}$), sp24 ($7.06 \times 10^{-6}$) and sp25 ($2.91 \times 10^{-8}$). The *e* value for the entire array, as opposed to that of each spacer, can be approximated multiplying the *e* value by the number of spacers. Low *e* values represent approximately the probability of obtaining a match which is better or equal. Therefore, the probability of obtaining four equal or better matches is roughly the product of their four *e* values. Under these considerations, the probability of obtaining this or better matches to the array of 111 spacers is approximately $111^4 \times 0.002 \times 7.48 \times 10^{-7} \times 7.06 \times 10^{-6} \times 2.91 \times 10^{-8}$, that is, $5 \times 10^{-14}$. Hence, there is a strong statistical correlation for the predicted virus–host pair.

**Sequences in this work**. All sequences in this study were downloaded from https://www.ncbi.nlm.nih.gov/nucletide/. The Methanomassiliicoccales sequences are listed in Supplementary Table 2, and their corresponding CRISPR sequences are summarized in Supplementary Table 3. A previously published set[11] of smacovirus genomic sequences was retrieved from accessions and is listed in Supplementary Table 4. An early reported set of CRESS-DNA sequences[2] was also retrieved from their accessions (no nucleotide sequence for Q80GM6 or P0C647 was found). Coding sequences were downloaded from the same sources of

sequence sets after selecting the appropriate option. Human coding sequences were downloaded from ftp://ftp.ncbi.nih.gov/pub/CCDS/current_human/CCDS_nucleotide.current.fna.gz.

**Additional similarity searches**. As the databases size increases, sensitivity is lost, as larger datasets imply a higher chance of alignments arising by chance. Therefore, to disclose potential matches that could have been overlooked, additional searches were performed with smaller sets of related sequences. Queries were spacers of *Ca.* M. intestinalis, while subjects were a set of smacoviruses. For a finer detection of spacer targets, non-seeded alignments were performed. A dynamic programming algorithm for sequence comparison was developed (referred to as local–global alignment) that combines global[56] and local[57] alignments, so that all length of query sequences are aligned to a local region of the subject. In the local–global alignment algorithm, the matrix cells that represent the first or last nucleotide of a query are processed with the local alignment algorithm, and the rest of the cells are processed with a global alignment algorithm; alignment traceback starts in the cells representing the last nucleotide of the query which comply with the scoring threshold. The parameters were as follows: match:+1, mismatch:−1, gap open:−2 and gap extend:−1. Specific *e* values for small sequences were calculated from alignments with random sequences for each query length. The cut-off *e* value used was 0.01. An additional search with the same parameters but using local alignments was performed to detect similarities to additional spacers that could correspond to sequences which have undergone a higher divergence. Protospacers to four additional spacers were detected. These local alignments are mentioned in a few occasions but were generally not taken into consideration. Matches to sp22 are an exception to the cut-off value (0.015) but were included because a BLASTn search with default parameters for short sequences recognized a shorter alignment in the same regions with an *e* value of 0.001. The discrepancy in *e* value and length of alignment is mostly due to a higher penalization of mismatches and gaps in BLASTn. The redundancy of subject sequences in the case of *Smacoviridae* (confirmed by hits of spacers in multiple genomes) implies that the expected values are an overestimation and, consequently, the probability of obtaining matches for a given spacer is lower than the *e* value indicated. In non-redundant subject sequences, a set of 111 queries with a threshold e-value of 0.01 is expected to randomly produce 0.01 ×111 (1.11) matches. However, redundancy and the presence of PAM sequences (see related section) make nearly all of the matches in local–global alignments believable, except for spacer 79, the only dubious match. Moreover, local–global searches were performed for all spacers found in Methanomassiliicoccales against smacoviruses and other CRESS-DNA viruses (datasets described above).

**PAM detection**. To identify PAM sequences, only representative sequences of best local–global matches in smacoviruses to 23 *Ca.* M. intestinalis spacers were selected (shown in Supplementary Table 1), that is, 10nt of each flanking region (upstream and downstream with respect to the spacer). A stack of each set of flanking sequences was used to create sequence logos[58] with the Weblogo program[59] from the website https://weblogo.berkeley.edu/logo.cgi.

**Design of an archetype guest**. The visualization of targets to *Ca.* M. intestinalis spacers in the genomes of smacoviruses cannot be achieved using a single genome, as the most targeted genome (KP264968) only contains 14 of the 23 targets. All smacoviruses share the same genomic architecture, which facilitates the design of an archetype virus to represent the positions of all protospacers, as shown in Fig. 1 at the bottom. The archetype virus was devised by identifying the positions of representative protospacers relative to the gene or intergenic region where they are located. The positions of the protospacers in the archetype are relative to their positions in the actual virus, for example, two-thirds from the start codon to the end. The lengths of genes and intergenic regions in the archetype represent the mean of the viruses containing the protospacers selected as best target representatives (KC545226, KJ577818, KM573770, KP233174, KP233175, KP233178, KP233179, KP233181, KP233184, KP233186, KP264966, KP264968, KY086298). The complementary sequence of KM573770 and KY086298 was used instead, to concur with the rest. CRESS-DNA viruses have a conserved nonanucleotide motif in the loop where replication starts. For instance, 5′-NANTATTAC-3′ is present in the viral sense strand of related viruses[10]. In smacoviruses, the loop is located near the end of the *rep* gene[1]. The nonanucleotide 5′-NAGTRTTAC-3′[11] was found shortly downstream of the *rep* stop codon, except in genomes with the opposite orientation (KM573770 and KY086298), which contain the *rep* termination codon in the nonanucleotide (5′-TAGTGTTAC-3′; antisense stop codon is underlined). Therefore, the chosen orientation theoretically reflects the viral ssDNA for all sequences.

**Stop codon analysis**. Methanogens such as *Ca.* M. intestinalis use the *amber* stop codon to encode for the amino acid pyrrolysine; therefore, the use of the *amber* as a termination codon is diminished in *Ca.* M. intestinalis[24]. To compare stop codon usage between smacoviruses and expected hosts, a Bio-Python[60] script was used. The script extracts the last three nucleotides of coding sequences and counts the appearances of each standard termination codon (*amber*: 'AUG'; *opal*: 'AGU'; *ochre*:

'UAA'). The percentage of each codon was then calculated (Fig. 3). The number of stop codons considered in each set was: Humans $n = 32553$, Ca. M. intestinalis $n = 1824$, CRESS-DNA $n = 1710$ and Smacoviridae $n = 188$.

**Reporting summary**. Further information on experimental design is available in the Nature Research Reporting Summary linked to this article.

**Code availability**. The programme performing local–global alignments could be made available upon request for specific purposes.

## Data availability

No new data was generated. All sequences used in this work can be downloaded from public databases as described in the Methods section 'Sequences in this work'.

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

## Acknowledgements

F.R.-V. was supported by grant 'VIREVO' CGL2016-76273-P [AEI/FEDER, EU] (cofounded with FEDER funds); Acciones de dinamización 'REDES DE EXCELENCIA' CONSOLIDER-CGL2015-71523-REDC from the Spanish Ministerio de Economía, Industria y Competitividad and PROMETEO II/2014/012 'AQUAMET' from Generalitat Valenciana. We thank Dr. Felipe Hernandes Coutinho for providing us with access to his sequence collection and metadata, which facilitated the initial search.

## Author contributions

C.D.-V. designed the study, programmed the code, performed analyses, interpreted results and wrote the paper. F.R.-V. provided feedback and revised the manuscript.

## Additional information

**Competing interests:** The authors declare no competing interests.

