## [Peer Review File · Nature Communications]

Reviewers' comments:

Reviewer #1 (Remarks to the Author):

In this provocative manuscript, the authors argue that uncultivable ssDNA viruses of the Smacoviridae family do not infect eukaryotic hosts, as has been assumed until now, but rather infect methanogenic archaea. Indeed, smacoviruses have been isolated from feces of various animals, but their actual hosts have not been identified. The claim that smacoviruses infect archaea is based solely on the match between CRISPR spacers of one methanogenic archaeal species and several closely related smacoviruses.

Spacer matches to the smacovirus genomes appear to be significant and can be detected reproducibly. This is certainly intriguing, but I do not find this observation to be sufficient or conclusive. In my opinion, more direct evidence of host association is necessary to claim the archaeal association. The following points raise my suspicion. (i) The 2.5 kb genome encoding only 2 genes (for a capsid and a Rep protein) appears insufficient to accomplish the infection of a prokaryotic cell. The authors rightly point out that all other prokaryotic viruses have larger genomes and encode more genes. There is a good reason for this. To specifically recognize their hosts and penetrate through the cell envelope viruses typically encode dedicated proteins other than the main capsid proteins. Furthermore, for virion release prokaryotic viruses also encode specialized protein. The claim of the authors, that *Methanomassiliicoccus intestinalis* might not have a cell envelope although *M. luminensis* was shown to have one, and that this will “allow traffic of small virus sized particles into their cytoplasm” (L248) is not substantiated. There are no such mechanisms known in prokaryotes.

(ii) In different *Methanomassiliicoccus* isolates, spacers against smacoviruses are present in only one archaeal strain, whereas other related members of the same order lack such spacers. Such an isolated occurrence of a single CRISPR array with a substantial fraction of “spacers” to closely related species of smacoviruses sequenced from the same source (human feces) casts doubt on the validity of the *M. intestinalis* genome assembly. For instance, it is not excluded that sequence of this particular archaeal genome has been wrongly cross-assembled from reads derived from the actual organism and from smacoviruses (i.e., with fragments of the viral genomes artificially ending up in the CRISPR array). Genome sequencing of additional *M. intestinalis* isolates (and preferably those of more distantly related species) could be undertaken to rule out this possibility. Alternatively, smacoviruses might not infect *M. intestinalis*, but their genomes might be non-selectively internalized, degraded and integrated into CRISPR arrays. In such case, it is also cannot be claimed that smacoviruses actually INFECT archaea. The only way to demonstrate infection is to actually demonstrate replication of smacovirus genomes in archaeal cells. This can be achieved, for instance, by electroporating the viral genome into *M. intestinalis* cells or by performing something similar to phageFISH (PMID: 23489642). Extraordinary claims should be supported by robust data.

Reviewer #2 (Remarks to the Author):

Diez-Villaseñor and Rodriguez-Valera report here two results, a strong one which is interesting for a wide readership, namely that smacoviruses infect Archaea, and a weak one, dealing with the possibility that the CRISPR-CasI-B system of *Ca Methanomassiliicoccus intestinalis* targets single strand DNA.

The first result is well substantiated and straight forward. The second one relies on the fact that two thirds of the spacers matching smacoviruses (ie 15 of 23 total) in this particular Archaeon are oriented such that they could target the single strand replicative intermediate of this rolling circle virus (pval 10%).

Given the interest of the first result, I would recommend publication, but after a thorough revision of the manuscript, along the lines indicated below.

Major points

1. Tune down and reduce to a few lines the observation dealing with the spacer strand bias and its putative meaning with respect to a single strand target of this CRIPR-Cas system, or prove it experimentally. This point is just the beginning of a story, and weakens the rest of the paper.
2. Restructure the manuscript along its strong axis, and give more biological background. For instance, now that the target of smacoviruses is found to be Archaea, the reader has more questions on other ssDNA viruses: nine families of ssDNA viruses among 13 are said to “infect or are associated with” eukaryotes: be more specific, and mention how many have proven eukaryotic host, and do the same among the CRESS: could it be that all CRESS viruses infect Archaea ?
3. Discuss better the question of archaeal cell wall crossing by smacoviruses (l. 243-249): arguments are all mixed up (Thermoplasmata lack a cell wall, but the Thermoplasmata *M. luminyensis* has a cell wall, and the *Ca. M. intestinalis* may have a weak cell wall): cannot a genome examination for the genes involved in cell wall building help decide on this (important) point, or an EM picture ?
4. The link between smacoviruses and disease (l. 249-257). Explain better why this could be an indirect link, not causative. Could not *Ca. M. intestinalis* occupy the niche once its natural inhabitants have been displaced by a pathogen, which still needs to be identified ? What is the prevalence of *Ca. M. intestinalis* in healthy human or animal subjects ?
5. The end of the discussion (l. 281-304) is questionable. If nothing particular remains to be discussed, make the manuscript shorter, with a single ‘results and discussion’ section.

6. The manuscript is hard to read, full of enigmatic sentences, comma are missing, the phrasing is often confusing and one has to guess what is meant (a list would be too long to make). It should be completely rewritten with help of a native English speaker.
7. The interest of Tables 1, S3 and S5 is not evident.
8. Explain better and give access to the program making local/global alignments.
9. L. 323: E-value are not “approximately probabilities”. Authors probably know very well that E-values depend on database size, which makes them very delicate to handle.

Reviewer #3 (Remarks to the Author):

Villasenor et al. present evidence that certain viruses in the Smacoviridae family, canonically classified as eukaryotic viruses, may actually infect prokaryotic hosts.

Compelling evidence includes (1) spacer matches with smacoviridae viral sequences and (2) the content of amber stop codons coding for pyrrolysine in these viral genomes, which the authors argue is indicative of a methanogen host.

In the text, the authors state that the system found in their host strain is a I-B system with a I-A cas8 gene rather than I-B cas8 gene. Whether this system is active is not explored and if literature on the subject exists, it is not cited. Additionally, it would be valuable to show if certain spacer mismatches allow for robust CRISPR targeting in this system, since the spacers described contain differences with the proposed targeted protospacer sequences.

While the spacer evidence presented is compelling, the work would benefit from a functional assay of some sort, that demonstrates that this system is indeed functional. This could include northern blotting for mature crRNA or RT-qPCR for expression of this system in culture. Assuming no genetic tools are available in the organism itself, another option is reconstitution of the system in *E. coli* or in another tractable microbe, where dsDNA or ssDNA targeting could be assessed. Lastly, in vitro reconstitution is also possible, as has been done previously for other type I systems. As the paper reads now, it is comprised mostly of a single significant observation of spacer matches (corroborated by the amber codon observation), but no other support or novelty is provided, which feels needed to this reviewer.

-The strand bias that is presented is worthy of note but perhaps the last line of the abstract is misleading, as there are clearly spacers on both strands and the low sample size might make this conclusion tenuous.

In addition, many ideas in the text are not clearly presented, which hampers the read-ability of the manuscript and the introduction seems a bit long and disjointed.

Examples:

-The termination codon data appear in the same paragraph as the discussion about the PAM. The relationship is not obvious and if they are indeed independent thoughts, this should be rewritten.

-the sentence "A comparison of the predicted host with described ones" does not explain what the "predicted host" versus "the described ones" refers to.

-the sentence "...in diverse genomes of smacoviruses a bias to target the viral strand was observed" does not delineate what the "viral strand" describes.

-the sentence "There were just 5 targets present in only one genome (1, 35, 48, 56, 79), being three of them (35, 48, 56) in KY086298 (chicken associated smacovirus)" leaves the reader unclear on which genome is being targeted.

-I also recommend (1) avoiding non-standard nomenclature where appropriate and (2) utilizing standard CRISPR-Cas terminology wherever possible. For example, viral DNA in a bacterial cell should not be referred to as "guest" DNA.

-the sentence "A sequence logo revealed PAM 'CCN' (Figure 3 a,c) located upstream of spacers" should be changed to "protospacer", since PAMs exist in targeted sequences rather than in CRISPR arrays and the fact that the 'CCN' PAM has been described for "other CRISPR sequences (ref. 19)" is only really helpful if this applies to I-B or I-A systems, not just a PAM that generally exists, as it is so short.

-The acronym for CRISPR should be amended to "Clustered Regularly Interspaced Short Palindromic Repeats" in the introduction.

It would be of benefit to the authors to have the manuscript edited by a colleague outside of the field and for whom English is a primary language to enhance clarity. As is, the study is incomplete and not ready for publication.

Reviewer #1 (Remarks to the Author):

In this provocative manuscript, the authors argue that uncultivable ssDNA viruses of the Smacoviridae family do not infect eukaryotic hosts, as has been assumed until now, but rather infect methanogenic archaea. Indeed, smacoviruses have been isolated from feces of various animals, but their actual hosts have not been identified. The claim that smacoviruses infect archaea is based solely on the match between CRISPR spacers of one methanogenic archaeal species and several closely related smacoviruses.

Spacer matches to the smacovirus genomes appear to be significant and can be detected reproducibly. This is certainly intriguing, but I do not find this observation to be sufficient or conclusive. In my opinion, more direct evidence of host association is necessary to claim the archaeal association. The following points raise my suspicion. (i) The 2.5 kb genome encoding only 2 genes (for a capsid and a Rep protein) appears insufficient to accomplish the infection of a prokaryotic cell. The authors rightly point out that all other prokaryotic viruses have larger genomes and encode more genes. There is a good reason for this. To specifically recognize their hosts and penetrate through the cell envelope viruses typically encode dedicated proteins other than the main capsid proteins. Furthermore, for virion release prokaryotic viruses also encode specialized protein. The claim of the authors, that Methanomassiliicoccus intestinalis might not have a cell envelope although M. luminensis

was shown to have one, and that this will “allow traffic of small virus sized particles into their cytoplasm” (L248) is not substantiated. There are no such mechanisms known in prokaryotes.

The presence of CRISPR spacers matching at high similarity a viral genome is taken as enough proof to support the association host to virus. How else could the sequence from the virus ends in a precise ID mechanism such as CRISPR? We are aware of the necessity of additional genes for successful infection of most prokaryotic hosts. However, the biology of *Ca. M. intestinalis* is poorly known and a 'shortcut' could be available. Regarding cell structure, the updated view is that *M. luminyensis* and other studied Methanomassilicoccales (*Ca. Methanoplasma termitum*, which is less related to *Ca. M. intestinalis*) have a double membrane with no cell wall or S-layer. In the case of *M. luminyensis* for example, the cell structure is easily disrupted, and cells lyse in distilled water. So, it should not be as difficult to penetrate as other prokaryotes.

-(ii) In different Methanomassiliicoccus isolates, spacers against smacoviruses are present in only one archaeal strain, whereas other related members of the same order lack such spacers. Such an isolated occurrence of a single CRISPR array with a substantial fraction of “spacers” to closely related species of smacoviruses sequenced from the same source (human feces) casts doubt on the validity of the M. intestinalis genome assembly. For instance, it is not excluded that sequence of this particular archaeal genome has been wrongly cross-assembled from reads derived from the actual organism and from smacoviruses (i.e., with fragments of the viral genomes artificially ending up in the CRISPR array). Genome sequencing of additional M. intestinalis isolates (and preferably those of more distantly related species) could be undertaken to rule out this possibility.”

Prokaryotic viruses that infect one single species, or even strain, are very frequent. Therefore, a lack of spacers in other species matching smacoviruses (with one mentioned exception) is not surprising at all. An assembly error resulting in a well-defined CRISPR array with contaminant DNA as spacers is an extremely unlikely event. Not only are all spacers of the same standard length, but also selected sequences should be flanked by sequences partially identical to the CRISPR sequence in order to be assembled, which is not the case. Also, cross-contamination cannot explain the miss-assembly of sequences next to a PAM motif in the correct orientation. As stated in the paper, no single genome contains all protospacers. Therefore, the contamination would have to be a mix of different co-occurring smacovirus strains at the same time, which does not seem likely. In addition, the contaminant smacoviruses would have grown in the culture without a host, so contamination with another sample would be the only option. As the assembly comes from a culture that is not pure, it would be all but impossible that the only contamination assembled into spacers would belong to smacoviruses. It would also be extremely unlikely that the contamination incorporates spacers preferably near the leader end.

"Alternatively, smacoviruses might not infect M. intestinalis, but their genomes might be non-selectively internalized, degraded and integrated into CRISPR arrays. In such case, it is also cannot be claimed that smacoviruses actually INFECT archaea."

The presence of a high number of spacers in the array supports that contact is sustained in time. Acquisition is not so effective as to incorporate so many spacers, representing a high proportion, from the same viral family, internalized "non-selectively". In the rare occasions when this happens, the cell with the incorporation would have been outnumbered in the archaeal population, and most probably lost if there is not an advantage. And certainly, it would be impossible that so many acquisitions have persisted without impossible rates of non-specific DNA penetration and CRISPR adaptation.

"The only way to demonstrate infection is to actually demonstrate replication of smacovirus genomes in archaeal cells. This can be achieved, for instance, by electroporating the viral genome into M. intestinalis cells or by performing something similar to phageFISH (PMID: 23489642). Extraordinary claims should be supported by robust data."

The culture of methanogens requires specialized equipment. We do not have samples of *Ca. M. intestinalis* or smacoviruses at our disposal and do not know if they are available, certainly not in public collections. Note that only one strain of *Ca. M. intestinalis* has been described, and the CRISPR-Cas system has numerous spacers presumably avoiding infection and replication of smacoviruses. Therefore, infection is not expected. The possible use of viral strains without targets does not guarantee that the host is compatible. More complex experiments with genetically engineered smacoviruses or *Ca. M. intestinalis* would be needed and there are no known genetic tools that work in the organism.

Also, the spacers need to be double-stranded to be incorporated into the CRISPR array. Although some CRISPR-Cas systems synthesize the dsDNA spacer precursor from RNA, no Type-I

subtype, including the present one, has the necessary genes. Therefore, at least the first step required for viral replication is accomplished.

Exact matches to just one CRISPR spacer have a high success rate for host prediction. But this case is extraordinary due to the elevated number of matches which imply a robust association. The findings can only be explained in the light of a well established and consolidated relationship between *Ca. M. intestinalis* and smacoviruses.

Reviewer 2.

Reviewer #2 (Remarks to the Author):

*Diez-Villaseñor and Rodriguez-Valera report here two results, a strong one which is interesting for a wide readership, namely that smacoviruses infect Archaea, and a weak one, dealing with the possibility that the CRISPR-CasI-B system of *Ca Methanomassiliicoccus intestinalis* targets single strand DNA.*

The first result is well substantiated and straight forward. The second one relies on the fact that two thirds of the spacers matching smacoviruses (ie 15 of 23 total) in this particular Archaeon are oriented such that they could target the single strand replicative intermediate of this rolling circle virus (pval 10%).

A clarification may be needed here: single strand is not a replicative intermediate. Canonical rolling circle replication of double-stranded DNA bacterial plasmids uses a single-stranded intermediate later converted to double-stranded DNA. This is not the case of CRESS-DNA viruses, which are ssDNA that form a double-stranded DNA intermediate during replication.

Given the interest of the first result, I would recommend publication, but after a thorough revision of the manuscript, along the lines indicated below.

Major points

1. Tune down and reduce to a few lines the observation dealing with the spacer strand bias and its putative meaning with respect to a single strand target of this CRIPR-Cas system, or prove it experimentally. This point is just the beginning of a story, and weakens the rest of the paper.

The revised manuscript shows less emphasis on ssDNA targeting.

2. Restructure the manuscript along its strong axis, and give more biological background. For instance, now that the target of smacoviruses is found to be Archaea, the reader has more

questions on other ssDNA viruses: nine families of ssDNA viruses among 13 are said to “infect or are associated with” eukaryotes: be more specific, and mention how many have proven eukaryotic host, and do the same among the CRESS: could it be that all CRESS viruses infect Archaea ?

The introduction has been re-structured to put more emphasis on the biological significance of the finding: the discovery that the host of some smacoviruses is an archaeon. Overall, the viral aspect is much more developed in this version over the CRISPR side of the story. Now, the introduction starts by a more detailed description of the smacovirus and CRESS-DNA viruses in general. The reader has now a better idea of what is known about hosts of CRESS-DNA viruses.

*3. Discuss better the question of archaeal cell wall crossing by smacoviruses (l. 243-249): arguments are all mixed up (Thermoplasmata lack a cell wall, but the Thermoplasmata *M. luminyensis* has a cell wall, and the *Ca. M. intestinalis* may have a weak cell wall): cannot a genome examination for the genes involved in cell wall building help decide on this (important) point, or an EM picture ?*

The first report about *M. luminyensis* cellular structure is now considered an erroneous interpretation of the micrograph. The consensus states that there is no cell-wall or S-layer structure, even though *M. luminyensis* has the genes to synthesize the precursor of pseudomurein. The genetic pool for this structure is similar in *M. luminyensis* and *Ca. M. intestinalis*.

See answer to reviewer's 1 similar question.

*4. The link between smacoviruses and disease (l. 249-257). Explain better why this could be an indirect link, not causative. Could not *Ca. M. intestinalis* occupy the niche once its natural inhabitants have been displaced by a pathogen, which still needs to be identified? What is the prevalence of *Ca. M. intestinalis* in healthy human or animal subjects?*

More research would be needed to address these concerns. A relationship between *Smacoviridae* and intestinal disease is suspected but not certain. Further discussion on a mechanism of disease would be highly speculative given also the lack of knowledge about *Ca. M. intestinalis* biology. And that lack of knowledge may be due to a low prevalence or undetectable numbers. Even in published studies about methanogens from human gut mentions of *Ca. M. intestinalis* are rare.

It is not that *Ca. M. intestinalis* occupies the niche of a pathogen. For example, a proposed role of methanogens in intestinal disease is reducing H₂ concentration, allowing the growth of other organisms. An example of an indirect relationship is that, numbers of methanogens, are increased when the colon has some pathology. Methanol, or another toxic substrate of *Ca. M. intestinalis*, could also be the cause of the disease. The disease could be caused by the presence of *Ca. M. intestinalis* (revealed by smacoviruses), or the opposite, originated by the elimination of the archaeon. In all cases, pathology depends on a complex interaction of gut microbes that can vary in different individuals.

5. The end of the discussion (l. 281-304) is questionable. If nothing particular remains to be discussed, make the manuscript shorter, with a single 'results and discussion' section.

The section has been rewritten, but not eliminated.

6. The manuscript is hard to read, full of enigmatic sentences, comma are missing, the phrasing is often confusing and one has to guess what is meant (a list would be too long to make). It should be completely rewritten with help of a native English speaker.

The manuscript has been completely rewritten to meet these points, and send to a professional scientific editing service.

7. The interest of Tables 1, S3 and S5 is not evident.

Tables S3 and S5 have been eliminated. However, we find Table 1 of interest as the sequences may facilitate the work of scientist that want to take a closer look and also are an example of results produced by the local-global alignment algorithm.

8. Explain better and give access to the program making local/global alignments.

A better description of the algorithm is now included in Material and Methods:

" A dynamic programming algorithm for sequence comparison was developed (referred to here as local-global alignment; data not shown) that combines global⁴⁹ and local⁵⁰ alignments, so that all length of query sequences is aligned to a local region of the subject. In the local-global alignment algorithm, the matrix cells that represent the first or last nucleotide of a query are processed with the local alignment algorithm, and the rest of the cells are processed with a global alignment algorithm; alignment traceback starts in the cells representing the last nucleotide of the query which comply with the scoring threshold."

This is an in-house program and not completely necessary for reproducibility of the results. If the editor considers access to the software necessary we will make it available.

9. L. 323: E-value are not "approximately probabilities". Authors probably know very well that E-values depend on database size, which makes them very delicate to handle.

E-values indeed are not approximately probabilities. However, it can be shown that, when expected values are low, they approach the probability of obtaining at least one result.

The probability of randomly obtaining a given number of alignments with a stated e-value as threshold can be modelled with a Poisson distribution. The expected value is equivalent to the mean (parameter lambda of the distribution) and X the actual number of alignments obtained. The probability of obtaining at least one alignment, $P(X \geq 1)$, for a given e-value (λ) would be $1 - e^{-\lambda}$. The difference between λ and $P(X \geq 1)$ is 10% when $\lambda=0.2$, 5% when $\lambda=0.1$, 0.5% when $\lambda=0.01$ etc... In any case, λ is always bigger than $P(X \geq 1)$ and the use of e-values to

approximate probabilities is an overestimation that strengthens the case. The obtained probability indeed refers to the database size, which in this case is considerable, and it is clear that the result is really improbable by chance.

Reviewer #3 (Remarks to the Author):

Villasenor et al. present evidence that certain viruses in the Smacoviridae family, canonically classified as eukaryotic viruses, may actually infect prokaryotic hosts.

Compelling evidence includes (1) spacer matches with smacoviridae viral sequences and (2) the content of amber stop codons coding for pyrrolysine in these viral genomes, which the authors argue is indicative of a methanogen host.

In the text, the authors state that the system found in their host strain is a I-B system with a I-A cas8 gene rather than I-B cas8 gene. Whether this system is active is not explored and if literature on the subject exists, it is not cited. Additionally, it would be valuable to show if certain spacer mismatches allow for robust CRISPR targeting in this system, since the spacers described contain differences with the proposed targeted protospacer sequences.

We could not find any experimental analysis of this system in the literature. Single-stranded DNA viruses have high recombination and mutation rates, being almost as variable as ssRNA ones. It would be useful to meet his high variability with a system tolerant of many mismatches. However, the high number of targeting spacers could have been selected specifically to confront this variability.

While the spacer evidence presented is compelling, the work would benefit from a functional assay of some sort, that demonstrates that this system is indeed functional. This could include northern blotting for mature crRNA or RT-qPCR for expression of this system in culture. Assuming no genetic tools are available in the organism itself, another option is reconstitution of the system in E. coli or in another tractable microbe, where dsDNA or ssDNA targeting could be assessed. Lastly, in vitro reconstitution is also possible, as has been done previously for other type I systems. As the paper reads now, it is comprised mostly of a single significant observation of spacer matches (corroborated by the amber codon observation), but no other support or novelty is provided, which feels needed to this reviewer.

The paper emphasizes now the prediction that smacoviruses have an archaeal host. A failure demonstrating functionality at the present time would not contradict the acquisition of smacoviral sequences in the past. Although the paper would benefit from experimental assays, we think the results are already noteworthy. It shows for the first time that widespread CRESS-DNA viruses, which include many uncharacterized ones, can infect prokaryotes. Thus far, other thing that infection of eukaryotes was considered only a remote possibility. And notably, this is the identification of the smallest virus to infect prokaryotes.

-The strand bias that is presented is worthy of note but perhaps the last line of the abstract is

misleading, as there are clearly spacers on both strands and the low sample size might make this conclusion tenuous.

The last line of the abstract has been rewritten to communicate the result better:

"A probable target strand bias suggests that, in addition to double-stranded DNA, the CRISPR-Cas system can target ssDNA, which has not been previously observed."

In addition, many ideas in the text are not clearly presented, which hampers the read-ability of the manuscript and the introduction seems a bit long and disjointed.

The manuscript has been rewritten with the purpose of eliminating ambiguities and better presentation of the ideas. Most of the introduction has been rewritten to highlight the relevance and biological meaning of the virus-host pairing, ideas are now better connected, and text is shorter.

Examples:

-The termination codon data appear in the same paragraph as the discussion about the PAM. The relationship is not obvious and if they are indeed independent thoughts, this should be rewritten.

As now the main aspect of the paper is the virus-host pairing we think it is better to include both ideas in the same section about observations which support that smacoviruses are the origin of spacers.

-the sentence "A comparison of the predicted host with described ones" does not explain what the "predicted host" versus "the described ones" refers to.

The sentence and context have been improved to make it clearer:

"Prior to this study, a preliminary search of viral genomes for sequences similar to CRISPR spacers was performed (data not shown), which relates viruses and their hosts. For certain viruses a host was already described. A comparison of the CRISPR-predicted hosts with described hosts uncovered the following unanticipated result:"

-the sentence "...in diverse genomes of smacoviruses a bias to target the viral strand was observed" does not delineate what the "viral strand" describes.

The following sentence is now included in the paper:

"CRESS-DNA viruses enter the cell as ssDNA (this strand will be referred to as the 'viral strand')"

-the sentence "There were just 5 targets present in only one genome (1, 35, 48, 56, 79), being three of them (35, 48, 56) in KY086298 (chicken associated smacovirus)" leaves the reader unclear on which genome is being targeted.

Changed to:

" There were 6 spacers (sp1, sp35, sp36, sp48, sp56, and sp79) with targets present in only one of the viral genomes; three of the spacers (sp35, sp48, sp56) had targets in KY086298 (chicken-associated smacovirus), two of the spacers (sp1 and sp36) had targets in KP264968 (human-associated) and one spacer (sp79) had a target in KM573770 (camel-associated)."

-I also recommend (1) avoiding non-standard nomenclature where appropriate and (2) utilizing standard CRISPR-Cas terminology wherever possible. For example, viral DNA in a bacterial cell should not be referred to as "guest" DNA.

Terminology has been revised throughout the paper.

-the sentence "A sequence logo revealed PAM 'CCN' (Figure 3 a,c) located upstream of spacers" should be changed to "protospacer", since PAMs exist in targeted sequences rather than in CRISPR arrays and the fact that the 'CCN' PAM has been described for "other CRISPR sequences (ref. 19)" is only really helpful if this applies to I-B or I-A systems, not just a PAM that generally exists, as it is so short.

The context has been changed and the word 'spacers' removed :

"A sequence logo revealed the PAM 'CCN' (Figure 3a, c) located upstream (bases -3, -2 and -1)."

The statement about the "previously described" PAM has been eliminated.

-The acronym for CRISPR should be amended to "Clustered Regularly Interspaced Short Palindromic Repeats" in the introduction.

This has been addressed.

It would be of benefit to the authors to have the manuscript edited by a colleague outside of the field and for whom English is a primary language to enhance clarity. As is, the study is incomplete and not ready for publication.

The revised manuscript has improved the clarity of many points and being submitted to a professional scientific editing service. As a study about how CRISPR reveal the association between smacoviruses and the archaeon, we think it is complete and relevant.

Reviewers' comments:

Reviewer #1 (Remarks to the Author):

The matches between CRISPR spacers and smacovirus genomes do suggest that smacoviruses infect *Methanomassiliicoccus*. However, this remains an indirect evidence, not a proof. In the revised manuscript, the authors do not present any experimental data to substantiate their observation. At the very least, the authors have to (and I would insist on this) tone down definitive claims, such as “smacoviruses infect Archaea” in the title and elsewhere in the text to “might infect” or something similar.

Additional remarks

The Results section starts with “Prior to this study, a preliminary search of viral genomes for sequences similar to CRISPR spacers was performed (data not shown), which relates viruses and their hosts”. It is unclear who did this search – the authors or someone else? Why not shown? This has to be explained or reference provided.

Line 99-101: The fact that “the only two genes predicted to have similarities with viruses in the genome are located between arrays B1 and B2” does not at all suggest that they are “a remnant of a viral transfer of the CRISPR-Cas system”. The virus could integrate between the arrays or the solitary viral genes could be transferred horizontally between the arrays, or the genes might not be viral but rather cellular with viral homologues, etc, etc. The provided information does not allow to make such a claim. Delete the sentence.

Section “Groups of smacoviruses infecting *Ca. M. intestinalis* Issoire-Mx1” (again, change “infecting” to something less definitive): this section is very hard to follow and is not really conclusive. I would suggest removing it, but if the authors insist on retaining it, an explanatory figure would be very helpful.

The first sentence of the Main text contains too many bits of information and should be split into separate sentences. Also, the Introduction is peppered with references to CRISPR-Cas systems, whereas there are only 3-4 references on CRESS-DNA viruses. Add more references on other CRESS-DNA viruses, especially for lines 36-45.

Line 41: ref 1 is somehow randomly placed and is actually not really appropriate.

Line 204: “sequenced”, not “isolated”.

Reviewer #2 (Remarks to the Author):

The authors have answered my main points, the manuscript now reads well, and the ssDNA target issue is tuned down. I have no further restriction and despite the very narrow basis for the argument that smacovirus infect Archaea, given the interest of this finding, I recommend publication.

Reviewer #3 (Remarks to the Author):

Villasenor et al. present evidence that certain viruses in the Smacoviridae family, historically classified as eukaryotic viruses, may actually infect prokaryotic hosts.

The overall organization and clarity of the manuscript was improved on tremendously since the first submission and the re-structuring of the work is very effective.

A few items that still need to be addressed are:

Comment 1: The last sentence of the introduction reads, “This finding also reveals the identification of the smallest prokaryotic virus known to date.”

While the spacers are highly informative, in the absence of experimental work, this statement is more challenging, as the existence of spacers does not prove that it is a virus or at least an autonomous virus, in particular due to its minimal genome (i.e. not a helper phage, plasmid, etc). This statement should be re-written to reflect these ambiguities. Perhaps it is a viral-like element that piggy backs on a larger virus?

Comment 2: The “Validity of the virus-host prediction” section.

E-value may not be the best metric here, as we know more about CRISPR spacers and where identity is important. For example sp49, sp69, and sp32, which are described as 'local' alignments, do not provide much validation for these being putative targets, especially sequences with indels in the protospacer relative to the spacer.

It seems that these entries detract from the detection of perfect or near-perfect spacer matches, which appears more credible. I would suggest an abridged table that shows hits that pass a more stringent threshold informed not only by E value, but also by knowledge of CRISPR spacer-protospacer rules, i.e. excluding hits that are missing a fragment of the spacer, or with many indels. A full list could be shown in supplemental.

Including references to studies on "seed" sequences and escaper analysis would also be useful here. I would also recommend that the authors represent the spacer-protospacer relationship differently in table 1. It would be useful to see the full spacer shown with the alignment to the putative protospacer, instead of truncating the alignment, since specific sequence interactions that are missing are important regarding the validity of a spacer-protospacer match.

The PAM should be put into context with other studies on Type I-B CRISPR and the promiscuity seen in those PAM sequences. A couple papers have investigated the I-B system in this way. The references 26, 27 are not as useful (Type I-E PAM).

Comment 3: The "Distribution of targets and matching spacers" section.

In the line that reads, "As expected^{28,29}, the targeting spacers were predominantly among the most recent; no similarities were found for the last 41 spacers (out of 111), and most matching spacers were near the leader (14 of the 27 first spacers in B1 array were targeting smacoviruses).", "targeting spacers" should be changed to "putative targeting spacers" or "best-match spacers" since there is no experimental evidence that the spacers are truly targeting the virus.

Comment 4: The "Viral ssDNA could be a target" section should be removed.

The observation of strand bias is reasonable to point out, but the conclusion of ssDNA targeting is not strong here. One could interpret the data in the opposite manner actually, i.e. the existence of spacers targeting the non-genome strand proves that dsDNA is targeted. Otherwise, why would any spacers be selected for that target the genome when it only exists as dsDNA?

Reviewers' comments:

Reviewer #1 (Remarks to the Author):

The matches between CRISPR spacers and smacovirus genomes do suggest that smacoviruses infect Methanomassiliicoccus. However, this remains an indirect evidence, not a proof. In the revised manuscript, the authors do not present any experimental data to substantiate their observation. At the very least, the authors have to (and I would insist on this) tone down definitive claims, such as "smacoviruses infect Archaea" in the title and elsewhere in the text to "might infect" or something similar.

*This has been changed throughout the text. The title has been changed from 'CRISPR reveals...' to 'CRISPR suggests...'.
Deleted.*

Additional remarks

The Results section starts with "Prior to this study, a preliminary search of viral genomes for sequences similar to CRISPR spacers was performed (data not shown), which relates viruses and their hosts". It is unclear who did this search – the authors or someone else? Why not shown? This has to be explained or reference provided.

The results were not shown but the search was explained in the Methods section 'Previous BLASTn search'. The beginning of 'Results' section has been modified to make this clearer.

Line 99-101: The fact that "the only two genes predicted to have similarities with viruses in the genome are located between arrays B1 and B2" does not at all suggest that they are "a remnant of a viral transfer of the CRISPR-Cas system". The virus could integrate between the arrays or the solitary viral genes could be transferred horizontally between the arrays, or the genes might not be viral but rather cellular with viral homologues, etc, etc. The provided information does not allow to make such a claim. Delete the sentence.

Deleted.

Section "Groups of smacoviruses infecting Ca. M. intestinalis Issoire-Mx1" (again, change "infecting" to something less definitive): this section is very hard to follow and is not really conclusive. I would suggest removing it, but if the authors insist on retaining it, an explanatory figure would be very helpful.

The title has been changed to "putatively infecting". This section comments Figure 2, which can be considered an explanatory figure.

The section has been simplified to make it easier to follow.

The first sentence of the Main text contains too many bits of information and should be split into separate sentences.

Sentences have been separated.

Also, the Introduction is peppered with references to CRISPR-Cas systems, whereas there are only 3-4 references on CRESS-DNA viruses. Add more references on other CRESS-DNA viruses, especially for lines 36-45.

New references on other CRESS-DNA viruses have been included.

Line 41: ref 1 is somehow randomly placed and is actually not really appropriate.

The reference has been removed.

Line 204: "sequenced", not "isolated".

This has been changed.

Reviewer #2 (Remarks to the Author):

The authors have answered my main points, the manuscript now reads well, and the ssDNA target issue is tuned down. I have no further restriction and despite the very narrow basis for the argument that smacovirus infect Archaea, given the interest of this finding, I recommend publication.

Reviewer #3 (Remarks to the Author):

Villasenor et al. present evidence that certain viruses in the Smacoviridae family, historically classified as eukaryotic viruses, may actually infect prokaryotic hosts.

The overall organization and clarity of the manuscript was improved on tremendously since the first submission and the re-structuring of the work is very effective.

A few items that still need to be addressed are:

Comment 1: The last sentence of the introduction reads, "This finding also reveals the identification of the smallest prokaryotic virus known to date."

While the spacers are highly informative, in the absence of experimental work, this statement is more challenging, as the existence of spacers does not prove that it is a virus or at least an autonomous virus, in particular due to its minimal genome (i.e. not a helper phage, plasmid, etc). This statement should be re-written to reflect these ambiguities. Perhaps it is a viral-like element that piggy backs on a larger virus?

The sentence has been changed to 'Although infection and viral autonomy should be empirically confirmed, this finding could mean the identification of the smallest prokaryotic virus known to date.'

Comment 2: The "Validity of the virus-host prediction" section.

E-value may not be the best metric here, as we know more about CRISPR spacers and where identity is important. For example, sp49, sp69, and sp32, which are described as 'local' alignments, do not provide much validation for these being putative targets, especially sequences with indels in the protospacer relative to the spacer. It seems that these entries detract from the detection of perfect or near-perfect spacer matches, which appears more credible. I would suggest an abridged table that shows hits

that pass a more stringent threshold informed not only by E value, but also by knowledge of CRISPR spacer-protospacer rules, i.e. excluding hits that are missing a fragment of the spacer, or with many indels. A full list could be shown in supplemental.

It is necessary to differentiate between functional requirements and inferred origin of CRISPR-Cas interference. As a metaphor, in an archaeological excavation it is possible to find broken pieces of pottery that can no longer serve their function. In the same manner, mutations that prevent interference are not so relevant as to discard their origin. We are trying to reconstruct events that took place an indefinite time ago. In the case of the smacoviruses we are not dealing with the original sequences, but probably with descendants of their relatives. In addition, these viruses have elevated mutation rates. Lack of effective immunity for these spacers is foreseeable. The matches detected only in local alignments, which seem to be the main concern, are treated separately and with more caution in the text. Also, local alignments have been moved to the supplementary material.

Including references to studies on “seed” sequences and escaper analysis would also be useful here. I would also recommend that the authors represent the spacer-protospacer relationship differently in table 1. It would be useful to see the full spacer shown with the alignment to the putative protospacer, instead of truncating the alignment, since specific sequence interactions that are missing are important regarding the validity of a spacer-protospacer match.

We analyzed the probability that a spacer/putative target mismatch appears in a nucleotide in the putative 'seed' sequence (according to PMID: 29649958; positions: 1-5,7-10, 12). Mismatches are indeed more likely for a given nucleotide if it is placed in the seed region (analysis of 12 spacers whose local-global alignments had at least one mismatch but no gaps). The probabilities of a mismatch are 0.18 in the seed position versus 0.13 in the rest. The chances that this bias or higher could be found randomly is 0.17. This could mean that there is a selection of mutations that hamper immunity, but it seems very weak and the way of calculation arbitrary. Also, the presence of targets with 100% spacer identity in the presumably targeted genomes makes that analysis unnecessary and it is more complicated to integrate both elements in a single interpretation. The mutations do not necessarily have to denote an 'escaping' pressure, as they are expected randomly. Again, the validity of interference is not the same as validity of spacer origin.

Table 1, which included local alignments has been moved to 'Supplementary Material'. For the 5 alignments found only in local alignments it makes no sense to extend the alignment as there is no similarity. Please, remember that local-global alignments include the whole length of the spacer (35-39bp).

The PAM should be put into context with other studies on Type I-B CRISPR and the promiscuity seen in those PAM sequences. A couple papers have investigated the I-B system in this way. The references 26, 27 are not as useful (Type I-E PAM).

The references have been changed to papers assessing PAM requirements in subtype I-B systems.

Comment 3: The “Distribution of targets and matching spacers” section.

In the line that reads, “As expected^{28,29}, the targeting spacers were predominantly among the most recent; no similarities were found for the last 41 spacers (out of 111), and most matching spacers were near the leader (14 of the 27 first spacers in B1 array were targeting smacoviruses).”, “targeting spacers” should be changed to “putative targeting spacers” or “best-match spacers” since there is no experimental evidence that the spacers are truly targeting the virus.

Changed as suggested.

Comment 4: The “Viral ssDNA could be a target” section should be removed.

The observation of strand bias is reasonable to point out, but the conclusion of ssDNA targeting is not strong here. One could interpret the data in the opposite manner actually, i.e. the existence of spacers targeting the non-genome strand proves that dsDNA is targeted. Otherwise, why would any spacers be selected for that target the genome when it only exists as dsDNA?

This is not actually the opposite interpretation, targeting of both ss and dsDNA are not mutually exclusive as has been recently described for Cas14 (PMID: 30337455). The title of the section has been removed and the text location changed, being the possibility of targeting the ssDNA mentioned but less advertised.

REVIEWERS' COMMENTS:

Reviewer #3 (Remarks to the Author):

I find that this paper is suitable for publication in Nature Communications.